# Cell Therapy with Human Reprogrammed CD8^+^ T-Cells Has Antimetastatic Effects on Lewis Lung Carcinoma in C57BL/6 Mice

**DOI:** 10.3390/ijms232415780

**Published:** 2022-12-12

**Authors:** Evgenii G. Skurikhin, Olga Pershina, Natalia Ermakova, Angelina Pakhomova, Mariia Zhukova, Edgar Pan, Lubov Sandrikina, Darius Widera, Lena Kogai, Nikolai Kushlinskii, Aslan Kubatiev, Sergey G. Morozov, Alexander Dygai

**Affiliations:** 1Laboratory of Regenerative Pharmacology, Goldberg ED Research Institute of Pharmacology and Regenerative Medicine, Tomsk National Research Medical Centre of the Russian Academy of Sciences, Lenin, 3, 634028 Tomsk, Russia; 2Ministry of Health of the Russian Federation, Siberian State Medical University, Moskovski, 2, 634050 Tomsk, Russia; 3Stem Cell Biology and Regenerative Medicine Group, School of Pharmacy, Whiteknights Campus, Reading RG6 6AP, UK; 4Blokhin National Medical Research Center of Oncology, 115522 Moscow, Russia; 5Institute of General Pathology and Pathophysiology, 125315 Moscow, Russia

**Keywords:** human reprogrammed CD8^+^ T-cells, Lewis lung carcinoma, C57BL/6 mice, xenotransplantation, antimetastatic activity

## Abstract

Using a model of Lewis lung carcinoma (LLC) in vitro and in vivo, we previously demonstrated increased antitumor activity in CD8^+^ T-cells reprogrammed with an MEK inhibitor and PD-1 blocker. In this follow-up study, we carried out the reprogramming of human CD8^+^ T-cells (hrT-cell) using the MEK inhibitor and PD-1 blocker and targeted LLC cells. The effects of hrT-cell therapy were studied in a mouse model of spontaneous metastasis of a solid LLC tumor. We found antimetastatic activity of hrT-cells, a decrease in the number of cancer cells and cancer stem cells in the lungs, and an increase in the number of T-cells in the blood (including effector T-cells). Thus, reprogramming of human CD8^+^ T-cells with an MEK inhibitor and PD-1 blocker with targeted training by tumor target cells is a potential platform for developing a new approach to targeted lung cancer therapy.

## 1. Introduction

Lung cancer is one of the most common forms of cancer and one of the leading causes of death in men and women. Non-small cell lung cancer (NSCLC) accounts for about 85% of all lung cancers. Depending on the histological subtype, the genetic profile of the tumor, and the stage of disease, NSCLC treatment includes surgery, radiation therapy, chemotherapy, as well as targeted therapies (Kirsten rat sarcoma virus (KRAS), epidermal growth factor receptor (EGFR), anaplastic lymphoma kinase (ALK) inhibitors). In ~25% of the cases, NSCLS can be successfully treated with radical surgical treatment. Nevertheless, the 5-year survival rates for NSCLC remain at merely 50–60% for stage I, and 30–40% for stage III NSCLC and above [1]. However, only 16% of lung cancer cases are diagnosed at an early stage [2]. In about half of all patients diagnosed with NSCLC, distant metastases are present [3]. This is one of the reasons for poor prognosis and low survival rates [4].

Despite significant progress achieved in NSCLC treatment, the search for new methods to increase the effectiveness of therapy remains an urgent scientific and medical challenge. An insufficient response to chemotherapy is associated with existing or acquired chemoresistance of the tumor [3]. Targeted therapy has improved survival and quality of life in patients with advanced NSCLC [5]. Low frequencies of target mutations in the EGFR and ALK genes in patients with NSCLC is the main reason for the lower response to targeted therapy. In addition, secondary mutations that occur during treatment contribute to the development of drug resistance [5].

CD8^+^ T-cells are the main effector cells involved in the antitumor response. Decreased cytotoxic activity of CD8^+^ T-cells in NSCLC is associated with an activation of the PD-1/PD-L1 signaling pathway [6]. This is believed to lead to the escape of tumor cells from the immune response [6]. There is a growing interest in the modification of T-cells. Currently, CAR T-cells are used in the treatment of leukemia. However, known examples of CART therapy for lung cancer have only limited antitumor activity [7,8]. This is associated with the immunosuppressive effects of the tumor, the tumor microenvironment, and the absence of tumor-specific antigens [7].

Existing lung cancer therapies predominantly target actively proliferating tumor cells. Meanwhile, it is known that recurrence, metastasis, and progression of lung cancer are associated with cancer stem cells (CSCs) [9,10]. Thus, it would be advantageous to specifically target CSCs. Currently, the identification of CSCs is challenging due to the lack of specific markers. Often, CD117, Axl, CD44, CD87, CD276, and EGFR are used to identify CSCs [11]. Sox2 is highly expressed in NSCLC [12], determines tumor chemoresistance, and contributes to maintaining tumor cell stemness [13].

Recent discoveries on the mechanisms of tumor development and resistance formation provide new opportunities for creating novel therapeutic approaches to the treatment of lung cancer. In this context, the use of checkpoint inhibitors in combination with chemotherapeutic agents has been shown to be effective in patients with NSCLC [14]. In line with this, we proposed an approach to reprogramming CD8^+^ T-cells based on the combined use of the MEK inhibitor (MEKi) and the PD-1/PD-L1 signaling pathway blocker nivolumab. In an orthotopic mouse model of Lewis lung carcinoma (LLC), cell therapy with reprogrammed CD8^+^ T-cells isolated from mouse bone marrow demonstrated strong antitumor effects [15]. We hypothesized that reprogramming of human CD8^+^ T-cells with obligatory targeted training by target tumor cells can be effective in the treatment of human lung cancer. Studies on animal models of diseases are intermediate and essential links in the development of cell therapy [16,17]. Consequently, results of xenotransplantation studies make it possible to take a step towards in vitro studies on human biomaterial and subsequent clinical studies.

In the present study, we assessed antitumor and antimetastatic effects of cell therapy with reprogrammed human CD8^+^ T-cells in a model of spontaneous metastasis of a solid LLC tumor in C57BL/6 mice.

## 2. Results

### 2.1. Morphological and Histological Analysis of the Lungs of Mice with Lewis Lung Carcinoma after Cell Therapy with Human Reprogrammed CD8^+^ T-Cells

#### 2.1.1. Histological Examination

On the 17th day after the injection of LLC cells, multiple metastases formed in the lungs of mice. Metastases were tumor nodes consisting of heterogeneous atypical cells, predominantly with hyperchromic nuclei. Cellular and nuclear polymorphism, anisochromia, and anisocytosis were noted, and giant multinucleated cells were found. A large number of nucleoli were detected in the nuclei of tumor cells. Multiple mitoses were observed. The nodules grew under the pleura, as well as in the lung parenchyma, peribronchially and perivascularly (Figure 1b,e). Edema and focal infiltration by macrophages and lymphocytes, germination of carcinoma into the lumen of vessels with the formation of tumor emboli were detected in the lung parenchyma. Metastases were well vascularized. In large metastases, foci of necrosis with perifocal neutrophilic inflammation were observed.

The histological picture of the lungs of LLC mice treated with hrT-cell was similar to that described above in untreated LLC mice. In both groups, multiple metastases in the lung tissue, reactive edema and inflammatory infiltration of the lung parenchyma, vascular hyperemia, and tumor emboli in the vessels were found (Figure 1). There were no differences in the histological picture of the lungs between these groups in terms of the prevalence of the tumor process, the intensity of inflammation, and the reaction of the lung tissue.

#### 2.1.2. Tumor Growth and Metastases

Cell therapy with hrT-cell did not affect the tumor mass, weight factor, or tumor volume in LLC mice (experimental group) compared to LLC mice without treatment (control group) on the 17th day of the experiment (Table 1 and Table 2).

On the 17th day of the experiment, all animals of the experimental and control groups had metastases in the lungs (Figure 1b). However, in the animals of the experimental group, the degree of lung damage by metastases was lower (by 27%) (3.42 ± 0.48—mice with LLC; 2.5 ± 0.45—mice with LLC + hrT-cells) than in the animals of the control group. Additionally, in the experimental group, there was a decrease in the average number of metastases (by 39.8%) (15.92 ± 2.42—mice with LLC; 9.58 ± 2.86—mice with LLC + hrT-cells) and the average weight of the lungs affected by metastases was lower (by 20%) (210.1 ± 5.59—mice with LLC; 168.1 ± 10.44—mice with LLC + hrT-cells).

### 2.2. Effect of Human Reprogrammed CD8^+^ T-Cells on Lung Cancer Cells and CSCs Isolated from Mice with Lewis Lung Carcinoma

Our results confirm that CD8^+^ T-cells, when treated with MEKi and nivolumab in vitro, become T-cells with high, stable CCR7 expression, are resistant to the cytotoxic effect of cancer cells, and are distinct from naive T-cells (Appendix A).

In lung cancers, different CSC populations have been identified according to the expression of some markers, such as CD44, CD90, CD117, EGF, Axl, and Sox2, in different combinations [15]. An attempt has already been made to study the interaction between CSCs and CD8^+^ T-cells [18]. We suggest that lung cancer cells and CSCs are potential targets for reprogrammed T-cells. On the 17th day of the experiment, we observed an increase in the number of cancer cells and CSCs in the lungs of C57BL/6 mice from the experimental group. This were proliferating cells (CD117^+^EGF^+^CD44^+^Sox2^+^, CD117^+^CD44^+^Sox2^+^, CD117^+^EGF^+^Sox2^+^, Axl^+^Sox2^+^, Axl^+^CD90^+^CD44^+^Sox2^+^, CD90^+^CD44^+^Sox2^+^, CD90^+^Sox2^+^, EGF^+^Sox2^+^, EGF^+^CD44^+^Sox2^+^, CD276^+^Sox2^+^, CD44^+^Sox2^+^, and CD117^+^Sox2^+^) (Figure 2) and cells with immunophenotypes Axl^+^, Axl^+^CD117^+^, CD45^-^Axl^+^, CD45^+^Axl^+^, and CD45^-^CD117^+^ (Figure 3). Injections of hrT-cells significantly reduced the percentage of tumor cells and CSCs in the lungs of LLC C57BL/6 mice compared to untreated LLC mice (d17) (Figure 2 and Figure 3, Appendix A). The exceptions were cells with immunophenotypes Axl^+^ and CD45^-^CD117^+^, the content of which did not change under the influence of cell therapy, as well as cells with the immunophenotypes CD45^-^Axl^+^ and CD117^-^EGF^-^CD44^-^PD-L1^+^, where the number increased.

### 2.3. Effect of Human Reprogrammed CD8^+^ T-Cells on Blood T-Cell Populations in Mice with Lewis Lung Carcinoma

In the LLC metastatic model, we studied different populations of T-cells in the blood of mice on the 17th day of the experiment. We have shown that the number of T-cells in the blood of C57BL/6 mice with LLC was significantly lower than in intact control mice.

Administration of hrT-cells caused an increase in a significant number of T-cell populations (CD3^+^CD8^+^, CD3^+^CD4^-^CD8^+^, CD3^+^CD8^+^PD-L1^+^, CD3^+^CD8^+^PD-1^+^, CD3^+^CD8^+^CD276^hi^, CD8^+^CD62L^+^CD44^-^, CD8^+^CD62L^+^CD197^+^CD45RA^+^, CD45RA^+^CD95^+^CD197^+^, CD45RA^+^CD95^+^CD197^-^) in the blood of mice with LLC compared to untreated mice with LLC (Day 17) (Figure 4, Appendix A). At the same time, the content CD8^+^CD62L^+^CD197^+^CD95^hi^, CD8^+^CD62L^-^CD197^-^CD95^hi^, CD8^+^CD44^low^CD62L^hi^ T-cells did not change during treatment (Appendix A).

## 3. Discussion

The standards of care in lung cancer treatment are surgery, radiation therapy, and chemotherapy [19]. Due to the insufficient effectiveness of these methods, the search for new therapeutic targets and approaches to the treatment of lung cancer is a scientific and practical challenge of current medicine. In understanding the role of the immune system in cancer, immunotherapy is seen as a potential milestone in the search for a cure for lung cancer [7,20]. However, anti-cancer vaccines and cytokine therapy have not shown convincing efficacy in the treatment of lung cancer. For example, in clinical trials, cancer vaccines have not shown a positive effect on the survival of patients with NSCLC. On the other hand, conflicting data have been obtained on the effects of cytokine therapy on prognosis in lung cancer [21,22]. Immune checkpoint inhibitors are widely used in clinical practice. Thus, nivolumab, ipilimumab, and pembrolizumab are used in the complex therapy of lung cancer [21,22].

T-cells are the main cells of antitumor immunity [23]. A decrease in cytotoxic and proliferative activity, and death of T-cells, observed due to the effect of the tumor and tumor microenvironment, is one of the reasons for the escape of the tumor from immune surveillance [24]. A promising approach to increasing the effectiveness of antitumor therapy is to restore the cytotoxic activity of T-cells by reprogramming [15]. It is important to target reprogrammed cytotoxic T-cells to CSCs associated with tumor metastasis and recurrence.

Previously, we have shown that reprogramming with MEKi and anti-PD-1 monoclonal antibody nivolumab and training to LLC cells reduced apoptosis of mouse bone marrow CD8^+^ T-cells (mrT-cells) and increased their cytotoxic activity in relation to LLC cells in vitro. In a mouse orthotopic LLC model, mrT-cells exhibited antitumor and antimetastatic effects. The positive effects of mrT-cell therapy in vitro and in vivo were associated with an inhibitory effect on the target, CSCs [15]. We considered these positive results as a sufficient basis for studying the effectiveness of the presented reprogramming approach for human CD8^+^ T-cells. The antitumor and antimetastatic activity of the hrT-cell was evaluated in a metastatic model of LLC in C57BL/6 mice with induced immunosuppression. The choice for the mouse lung cancer model is due to the fact that LLC largely reproduces the pathogenesis of human NSCLC [25].

During histological examination of the lung tissue, we did not find significant differences between the animals of the control and experimental groups (Figure 1). Injection of hrT-cells also did not affect the mass and volume of the tumor in C57BL/6 mice of the metastatic model of LLC. Apparently, two injections of hrT-cells and a short observation period are not sufficient for significant changes in these tumor morphological parameters. At the same time, even a short course of treatment significantly reduced the degree of lung metastases, as indicated by a decrease in the average number of lung metastases and average lung weights.

As mentioned above, CSCs largely determine tumor heterogeneity, its survival, and development during treatment, and play an important role in lung cancer metastasis [26]. In this regard, we evaluated different populations of CSCs in the lungs. CSCs were identified using common markers such as CD117, CD44, EGF, and Axl, as well as the cell proliferation marker Sox2 [27]. Interestingly, we have found similar markers in the same combination when mrT-cells were used for cell therapy [15]. The results of the cytometric analysis of in the present study indicate a significant decrease in the number of tumor cells and CSCs in the lungs of mice of the metastatic model of LLC after hrT-cell injection (Figure 2). We do not rule out that hrT-cells are recruited to the lungs, where they produce cytotoxic effects. However, the effect of cell therapy did not extend to all subpopulations of CSCs. For example, the content of CSCs with the phenotype CD117^-^EGF^-^CD44^-^PD-L1^+^ and CD45^-^Axl^+^ in treated mice with LLC remained at the same high level as in untreated mice with LLC (Figure 3). This may be explained by the selective cytotoxic effect of hrT-cells on certain targets [28].

Next, we compared the effects of hrT-cells on tumor cells and CSCs in an LLC mouse model with those of mrT-cells. It turned out that hrT-cells, like mrT-cells, have an inhibitory effect on cells with phenotypes Axl^+^, Axl^+^CD117^+^, CD90^+^Sox2^+^, EGF^+^CD44^+^Sox2^+^, EGF^+^Sox2^+^, CD44^+^Sox2^+^, CD117^+^Sox2^+^, and CD117^+^EGF^+^CD44^+^Sox2^+^. The revealed differences in the cytotoxicity of hrT-cells and mrT-cells are apparently associated with species differences in reprogrammed cells.

CD8^+^ T-cells are the most powerful effectors in the anti-cancer immune response. After activation, CD8^+^ T-cells effectively lyse tumor cells in close proximity to them [18]. Reprogramming with the use of MEKi and PD-1 blocker contributed to a change in the properties of T-cells: a population of effector T-cells with the properties of memory cells was formed (T_SCM_) [15]. We hypothesized that the antitumor effect of hrT-cell therapy revealed in this study may be associated with an increase in the numbers of cytotoxic T-cells (Figure 3). In addition to the populations of T-cells studied in the previous work [15], we additionally evaluated the content of functional effector T-cells CD3^+^CD8^+^PD-L1^+^ [29] and terminally differentiated effector T-cells CD45RA^+^CD95^-^CD197^-^ [30] (Figure 4). After hrT-cell xenotransplantation, the content of CD8^+^CD62L^+^CD44^-^ cells, defined as minimally differentiated memory stem T-cells, increased in the blood [31]. At the same time, we found CD8^+^CD62L^hi^CD197^hi^CD95^+^ cells, which are defined as memory stem cells (T_SCM_) [32]. T_SCMs_ are capable of generating all three subsets of memory cells and T_EFF_ cells [33,34], and they are rare antigen-experienced T-cells with the ability for long-term self-renewal and multipotency and generated directly from naive lymphocytes [32].

CD3^+^CD8^+^ T-cells exist both as effector and naïve cells. According to our data, hrT-cell caused the replenishment of populations of naive T-cells (CD8^+^CD62L^+^CD197^+^CD45RA^+^; CD45RA^+^CD95^+^CD197^+^; CD8^+^CD197^+^) [34].

Additionally, we found an increase in CD3^+^CD8^+^CD279^hi^ T-cells that are capable of expressing CXCL13 and CD3^+^CD8^+^PD-1^+^ T-cells (Figure 4). We consider this as a positive effect. Thus, increased expression of CXCL13 in the tumor microenvironment attracts other subpopulations of immune cells with tumoricidal activity [35]. PD-1 is predominantly expressed in activated T-cells but is also expressed in T-cells depleted due to chronic antigen stimulation [36,37]. When hrT-cell therapy leads to a decrease in tumor progression and metastasis, it is more relevant to refer to an adaptive immune response and the generation of activated mouse CD3^+^CD8^+^PD-1^+^ T-cells in the metastatic mouse model of LLC.

It should be noted that in the present study, we observed significantly more populations of CD8^+^ T-cells involved in antitumor immunity than indicated in the results. However, the content of these cells did not change during xenotransplantation (Appendix A).

The obtained results indicate the possibility of a direct cytotoxic effect of hrT-cells on tumor cells and CSCs. However, the paracrine activity of hrT-cells and, as a consequence, an increase in the blood of mice with LLC of mouse T-cells (for example, naive, effector, with stem potential) cannot be excluded. This hypothesis is based on the assertion of various teams of authors that paracrine cell therapy can improve the autocrine regulation of subpopulations of T-cells and their progenitors [38,39]. Presumably, the additional antitumor and antimetastatic effects of hrT-cells may be due to the paracrine mechanism of the increase in the number of immune cells.

Thus, the results of this study demonstrate the effectiveness of our previously proposed scheme for the reprogramming of CD8^+^ T-cells. hrT-cells demonstrated antitumor and antimetastatic activity in vivo. The positive effect of hrT-cells on the metastatic model of LLC in immunosuppressed mice was associated with the elimination of tumor cells and CSCs.

Although murine LLC is comparable in many ways to human NSCLC, our study is limited by the capacity of animal models to mimic human lung cancer. Thus, further evaluation in more mouse experimental systems and randomized trials is needed.

## 4. Materials and Methods

### 4.1. Animals

Six-to-eight-week-old male C57BL/6 mice were obtained from the nursery of the Experimental Biological Models Department of the E. D. Goldberg Research Institute of Pharmacology and Regenerative Medicine (veterinary certificate is available). Animal keeping and design of experiments were approved by the Ethics Committee of the E. D. Goldberg Research Institute of Pharmacology and Regenerative Medicine (protocol No. 189092021 from 11.10.21). The animals were maintained in accordance with the European Convention for the Protection of Vertebrate (Strasbourg, 1986); Principles on Good Laboratory Practice (OECD, ENV/MC/CUEM (98)17, 1997).

### 4.2. Lewis Lung Carcinoma Cell Line

The Lewis lung carcinoma (LLC) cell line obtained from the C57BL mouse strain (400263 CLS Cell Lines. Service, GmbH, Eppelheim, Germany) was used in experiments in vivo and in vitro.

### 4.3. Lewis Lung Carcinoma Cell Culture

LLC cells were plated at a seeding density of 3 × 10^5^ cells/1 cm^2^ in T-25 flasks. The cells were maintained in RPMI 1640 medium (Sigma-Aldrich, St. Louis, MO, USA) supplemented with 2 mM L-glutamine (Sigma-Aldrich, St. Louis, MO, USA) and 10% fetal bovine serum (FBS, Sigma-Aldrich, St. Louis, MO, USA) at 37 °C in a humidified atmosphere containing 5% CO_2_. The culture medium was changed 2–3 times per week.

### 4.4. Modeling Immunosuppression in Mice

All animals were administered ketoconazole (Torrent pharmaceuticals, Indrad—382721, Dist. Mehsana, India) orally at a dose of 10 mg/kg in 0.5% solution of carboxymethylcellulose (Carboxymethylcellulose sodium salt, Sigma-Aldrich, St. Louis, MO, USA) and cyclosporine (TEVA Czech Industries, Opava, Czech Republic) at a dose of 30 mg/kg intraperitoneally for 7 days, from day 10 to day 4 prior to LLC injection. On days 3 and 1 prior to injection of LLC cells, cyclophosphamide was administered subcutaneously at a dose of 60 mg/kg (Baxter Oncology GmbH, Halle/Westfalen, Germany). To monitor the immunosuppression, white blood cell count (WBC) was calculated 1 day before the administration of the LLC cell suspension [40].

### 4.5. Metastatic Model of Lewis Lung Carcinoma

Mice were injected subcutaneously into the right axillary region with 5 × 10^6^ LLC cells suspended in 100 µL of RPMI 1640 medium (Sigma-Aldrich, St. Louis, MO, USA) [41]. Tumor growth was monitored every three days. For this purpose, the linear dimensions of the tumor nodes were measured in orthogonal planes in all animals and the volumes of the tumors were calculated in the elliptical approximation [42]. The administration of LLC cell suspension was set as day 0 of the experiment. 17 days post-LLC cell inoculation, the animals were euthanized. To assess the development of the tumor process, we measured the size of the tumor, and morphological and histological examination of the lungs.

### 4.6. Characteristics of a Healthy Volunteer

This study was a pilot investigation. A blood sample from 1 healthy volunteer was used. A volunteer K was selected from the general group of volunteers before [43]. Characteristics of the volunteer K (age (years)—35; sex—male; non-smoker; chronic obstructive pulmonary disease—without; lung cancer—negative). Informed consent was obtained from a healthy volunteer.

### 4.7. Study Design

The experimental design is shown in Figure 5. At stage 1, immunosuppression was modeled in C57BL/6 mice and a model of spontaneous metastasis of solid LLC tumors was induced. At stage 2, CD8^+^ T-cells isolated from the blood of a healthy volunteer were reprogrammed with targeted training by LLC cells. At stage 3, we evaluated the antitumor and antimetastatic activity of human reprogrammed CD8^+^ T-cells using the LLC model of spontaneous metastasis in C57BL/6 mice with induced immunosuppression. At the same time, the content of various populations of tumor cells and CSCs in the lungs and T-cells in the blood was studied. After modeling immunosuppression, the mice were separated into 2 groups: mice with LLC and mice with LLC treated with hrT-cells. A separate group consisted of intact mice (Intact mice).

### 4.8. Isolation of Human Blood Mononuclear Cells

Lympholyte-H (CEDARLANE, Netherlands, Cedarlane Laboratories, Cat#CL5015) protocol was used for the elimination of erythrocytes and dead cells from human blood as well for the isolation of mononuclear cells.

### 4.9. Cryopreservation of Mononuclear Cells Obtained from Human Blood

For cryopreservation of mononuclear cells obtained from the blood of a healthy volunteer, CryoStor^®^ CS5 cryopreservation medium (serum-free, containing 5% dimethyl sulfoxide (DMSO) StemCell Technologies, WA, USA) was used. Cold (2–8 °C) CryoStor^®^ CS5 was added to the cell suspension at a rate of 10 million cells/1 mL. The suspension was thoroughly mixed and placed in a cryovial. Cells were incubated at 2–8 °C for 10 min. The suspension was then cooled using a controlled slow-rate cooling protocol (1 °C/minute) and stored at liquid nitrogen temperature (−135 °C). Cells were thawed in a water bath at 37 °C and culture RPMI 1640 medium prewarmed to 37 °C and supplemented with 10% fetal bovine serum, 10 mM HEPES (Sigma-Aldrich, St. Louis, MO, USA), and 55 µM β-mercaptoethanol (Thermo Scientific™ 35602BID, Thermo Scientific, Waltham, MA, USA) in sample:culture medium = 1:10 (Sigma-Aldrich, St. Louis, MO, USA) (FBS, Sigma-Aldrich, St. Louis, MO, USA), was added. The suspension was washed twice at 300 g for 10 min at room temperature (15–25 °C).

### 4.10. Magnetic Separation of Human CD8^+^ T-Cells

Magnetic separation was performed to enrich the cell fraction with CD8^+^ T-cells. Enrichment was performed following a standard protocol using a human kit (EasySep^TM^ Human CD8^+^ T Cell Isolation Kit), as recommended by the manufacturer (Stemcell Technologies, Vancouver, BC, Canada).

### 4.11. Reprogramming of Human CD8^+^ T-Cells

After enrichment of the cell suspension with CD8^+^ T-cells using magnetic separation, the cells were incubated in the medium recommended for CD8^+^ T-cells (RPMI 1640 (Sigma-Aldrich, St. Louis, MO, USA) with the addition of 10% FBS (Sigma-Aldrich, St. Louis, MO, USA), 2 mM L-glutamine (Sigma-Aldrich, St. Louis, MO, USA), 10 mM HEPES (Sigma-Aldrich, St. Louis, MO, USA) and 55 μM β-mercaptoethanol (Thermo Scientific™ 35602BID, Thermo Scientific, Waltham, MA, USA), 37°C, 5% CO_2_) for 2–3 h. The concentration of T-cells was 1×10^8^/mL. The volume of the medium in the vial was at least 5 mL.

An antigen-presenting mix was prepared from LLC cells lysate by using a freeze-thaw cycle in 0.85% NaCl solution. The cycle was repeated five times in rapid succession from −70 °C to 37 °C, and then re-frozen and stored at −70 °C before use. After the final thawing, the lysate was stained by trypan blue (Sigma-Aldrich, St. Louis, MO, USA) as described earlier [15]. The preparation of the adjuvant (Freund’s adjuvant) for the antigen-presenting mix was carried out according to the manufacturer’s standard protocol (Sigma-Aldrich, St. Louis, MO, USA). Freund’s adjuvant solution was mixed with the tumor cell lysate (3 × 10^4^/mL) at a 1:1 ratio to form a thick emulsion.

Human CD8^+^ T-cells were reprogrammed as described earlier [15]. Monoclonal antibody nivolumab (Bristol-Myers Squibb Company, New York, NY, USA) and MEK1/2i (cat. PZ0162, Sigma-Aldrich, St. Louis, MO, USA) were used for reprogrammed. Immunophenotypes of reprogrammed CD8^+^ T-cells were analyzed using Cytation 5 (BioTek Instruments, Inc., Winooski, VT, USA) [15].

### 4.12. Detection of the CCR7 Expression and Cytotoxicity in CD8^+^ T-Cells In Vitro

Images of cells were obtained using the cell-imaging Cytation 5 (BioTek Instruments, Inc., Winooski, VT, USA) instrument equipped with the following light cubes: DAPI (blue), GFP (green), RFP (yellow).

To assess CCR7 expression, T-cells were stained with anti-CCR7 antibodies and polyclonal secondary antibody donkey anti-Rabbit IgG H&L Alexa Fluor^®^ 555 (all Abcam, Cambridge, MA, USA). Nuclei were additionally stained with Hoechst 34580 (blue); CD8 FITC (green) was used for CD8^+^ T-cell detection. The percentage of CD8^+^CCR7^+^ cells was determined as the ratio of cells counted in green and red channel to total cells counted using the blue (DAPI) channel.

All images were obtained with Cytation 5 (4× or 20× magnification) followed by cell analysis using Gen5™ data-analysis software (BioTek, Instruments, Friedrichshall, Germany). Prior to the analysis, images were preprocessed to align the background.

After in vitro reprogramming, the expression of the chemokine receptor CCR7 (CD197) hrT-cell was evaluated (Appendix A).

Cytotoxicity of CD8^+^ T-cells was studied in cell culture of LLC as described previously [15]. Cytotoxicity of CD8^+^ T-cells in LLC culture is assessed by analyzing the ratio of cells counted in the blue and red channels to the total number of LLC (percentage of dead Hoechst^+^7AAD^+^ LLC). Determination of the percentage of dyed T-cells Hoechst^+^7AAD^+^ is made by the ratio of cells counted in blue and red channels to total cells of LLC without green channel (Appendix A).

### 4.13. Injection of Reprogrammed CD8^+^ T-Cells

To assess the antitumor and antimetastatic activity, reprogrammed human CD8^+^ T-cells were administered intravenously to recipient mice with LLC at 1 × 10^6^ cells/mouse in 0.1 mL of PBS on the 14th and 16th days of the experiment (Figure 5).

### 4.14. Isolation of Mouse Mononuclear Cells

Mononuclear cells from blood and lungs were isolated as described previously [44,45].

### 4.15. Detection of the Cancer Cells and CSCs Receptor Expression on Mouse Blood and Lung Mononuclear Cells

Expression of membrane and intracellular receptors of CSCs and T-cells isolated from the blood and lungs of mice was studied using mouse monoclonal antibodies in accordance with standard flow cytometry protocols. Briefly, the cell suspension was pre-incubated for 5 min with anti-mouse CD16/CD32 (FcBlock™, BD Biosciences, San Jose, CA, USA). After pre-incubation, the cell suspension was stained with fluorophore-conjugated monoclonal antibodies CD3 PerCP, CD4 FITC, CD8 BV510, CD44 APC-Cy™7, CD45 PerCP, CD45RA PerCP-Cy™5.5, CD62L APC, CD90 APC, CD95 BV421, CD117 FITC, CD197 (CCR7) PE, CD274 (PD-L1) PE, CD276 FITC, CD279 (PD-1) BV421, EGF (F4/80) Alexa Fluor^®^ 647, and Axl BV421 (dilution 1/50, BD Biosciences, San Jose, CA, USA). Appropriate isotype controls were used. For staining with intracellular marker, the cell suspension was stained with the Sox2 PE intracellular antibody (1/50 dilution, BD Biosciences, Franklin Lakes, NJ, USA). For analysis, a FACSCanto II flow cytometer with FACSDiva software was used (BD Biosciences, Franklin Lakes, NJ, USA).

### 4.16. Histological Examination of the Lungs

Lung preparations were fixed in 10% neutral buffered formalin, passed through increasing concentrations of alcohol to xylene and embedded in paraffin wax according to standard methods, then sectioned into 5 μm-thick slices, and stained with hematoxylin and eosin [46].

### 4.17. Assessment of Tumor Growth

The effect of cell therapy on LLC growth was assessed by statistical comparison of the tumor node volume in the control and experimental groups at the different observation periods, according to the duration of tumor growth retardation and tumor growth inhibition index (TGII) [47]:TGII = (Vc − VO)/Ve × 100%, 
where Vc and Ve are the average node volume in the control and experimental groups.

### 4.18. Assessment of Tumor Volume

Linear dimensions of tumor nodes were measured in orthogonal planes and their volume was calculated in the elliptical approximation [48]. The volume of tumors was calculated according to the following formula:V = π/6 × length × width × height, 

Additionally, a weighting factor was determined. The weight coefficient was calculated as the ratio of the tumor weight in milligrams to the animal weight in grams.

### 4.19. Evaluation of the Severity of the Metastatic Process [49]

The severity of the metastatic process was evaluated by the following indicators:-The frequency of tumor metastasis is the percentage of animals with metastases in relation to the total number of animals in the group;-The degree of damage to the lungs by LLC metastases, where

0 degree—no metastases;

1 degree—the number of metastases is less than 10 and no more than 1 mm in diameter;

2 degree—the number of metastases from 10 to 30 metastatic nodes;

3 degree—the number of metastases is more than 30, metastases of various sizes;

4 degree—the number of metastases is less than 100, metastases without confluent growth;

5 degree—the number of metastases is more than 100 pieces, the presence of solid tumor nodes;

-Average number of metastases per animal in each group;-Is the average weight of the lungs affected by LLC metastases;-Metastasis inhibition index (MII)—MII was calculated by the formula:

MII = ((Ac × Bc) − (Ae × Be))/(Ae × Be) × 100%, 
where Ac and Ae are the frequency of lung metastasis in mice of the control and experimental groups, respectively, Bc and Be are the average number of lung metastases per animal in the control and experimental groups, respectively.

### 4.20. Statistical Analysis

Statistical analysis was performed by methods of variation statistics using the SPSS 12.0 software package (SPSS Inc., Chicago, IL, USA). The arithmetic mean (M), error of the mean (m), and probability value (*p*) were calculated. The distribution of a sample’s variable approximates a normal distribution was estimated using Skewness and Kurtosis measurements. Significance of differences was evaluated using the parametric Student’s t test or nonparametric Mann–Whitney U test. The difference between the two compared values was significant at *p* < 0.05.

## 5. Conclusions

In our study, we demonstrated that CD8^+^ T-cells isolated from a healthy volunteer reprogrammed by inhibition of the MAPK/ERK pathway via MEK1/2i, blockade of the PD-1/PD-L1 signaling pathway by the human monoclonal antibody nivolumab, and targeted training by LLC cells have antimetastatic activity in C57BL/6 mice in a metastatic model of LLC. The effect of hrT-cell therapy has been associated with an increase in cytotoxic T-cells in mice, and the effect of hrT-cells and mouse effector CD8^+^ T-cells on tumor cells and CSCs.

## Figures and Tables

**Figure 1 ijms-23-15780-f001:**
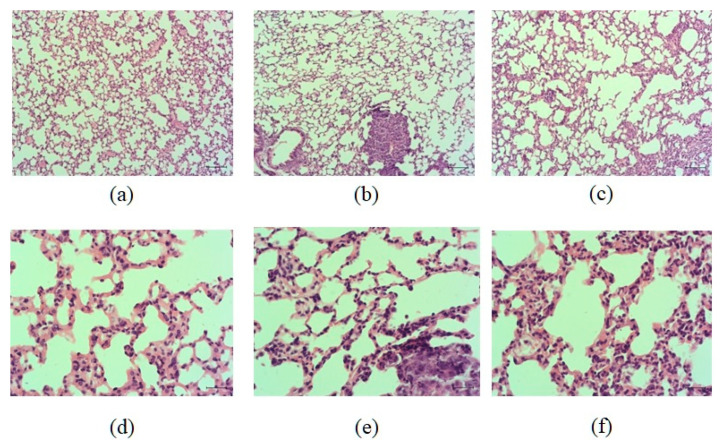
Micrographs of lung sections obtained from male C57BL/6 (**a**,**d**) mice of intact control; (**b**,**e**) mice with LLC; (**c**,**f**) mice with LLC treated with human reprogrammed CD8^+^ T-cells on d17. Tissues were stained with hematoxylin–eosin. ×100 (**a**–**c**) and ×400 (**d**–**f**). Scale bar 50 μm (**a**–**c**). Scale bar 10 μm (**d**–**f**).

**Figure 2 ijms-23-15780-f002:**
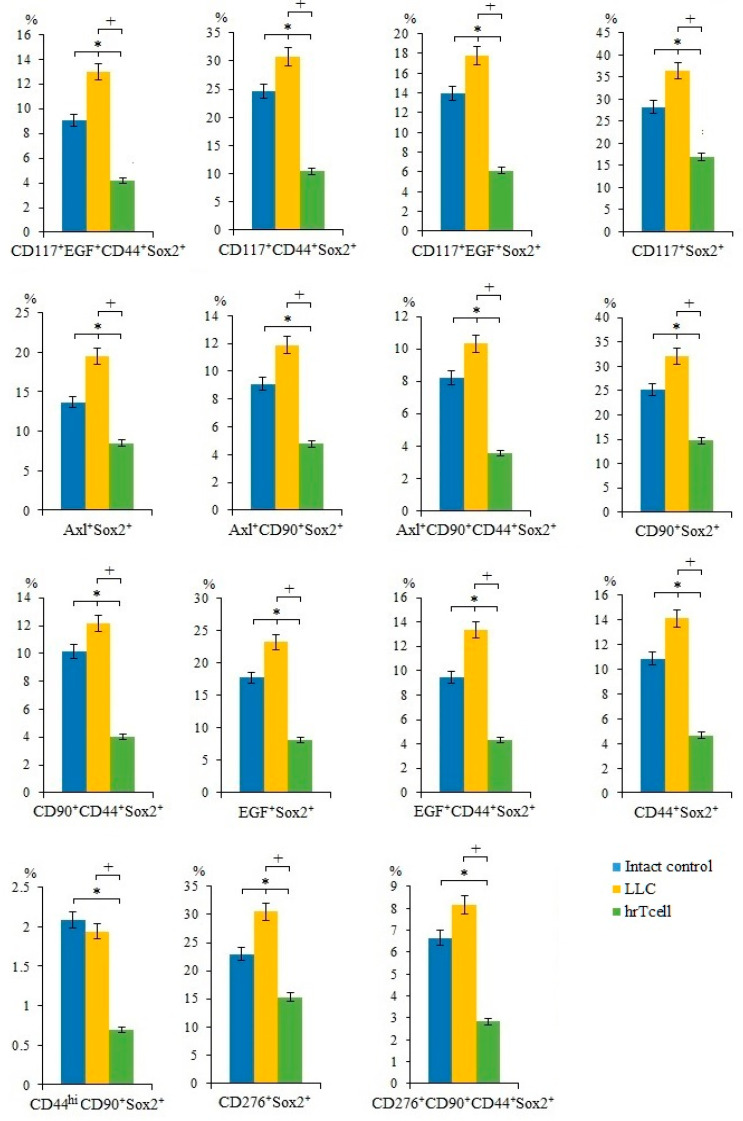
The effect of therapy with human reprogrammed CD8^+^ T-cells (hrT-cell) on the level of cancer cells and cancer stem cells (CSCs). The proportion (% of total mononuclear cells number) of CD117^+^EGF^+^CD44^+^Sox2^+^, CD117^+^CD44^+^Sox2^+^, CD117^+^EGF^+^Sox2^+^, CD117^+^Sox2^+^, Axl^+^Sox2^+^, Axl^+^CD90^+^Sox2^+^, Axl^+^CD90^+^CD44^+^Sox2^+^, CD90^+^Sox2^+^, CD90^+^CD44^+^Sox2^+^, EGF^+^Sox2^+^, EGF^+^CD44^+^Sox2^+^, CD44^+^Sox2^+^, CD44^hi^CD90^+^Sox2^+^, CD276^+^Sox2^+^, and CD276^+^CD90^+^CD44^+^Sox2^+^ CSCs in lungs of C57BL/6 mice with LLC on d17. Cells were analyzed by flow cytometry using antibodies for CD117, CD44, CD276, CD90, Axl, EGF, and Sox2. * Differences are significant in comparison with intact group (*p* < 0.05); +—for comparison with the mice with LLC (metastatic model). The results are presented from 3 independent series of experiments.

**Figure 3 ijms-23-15780-f003:**
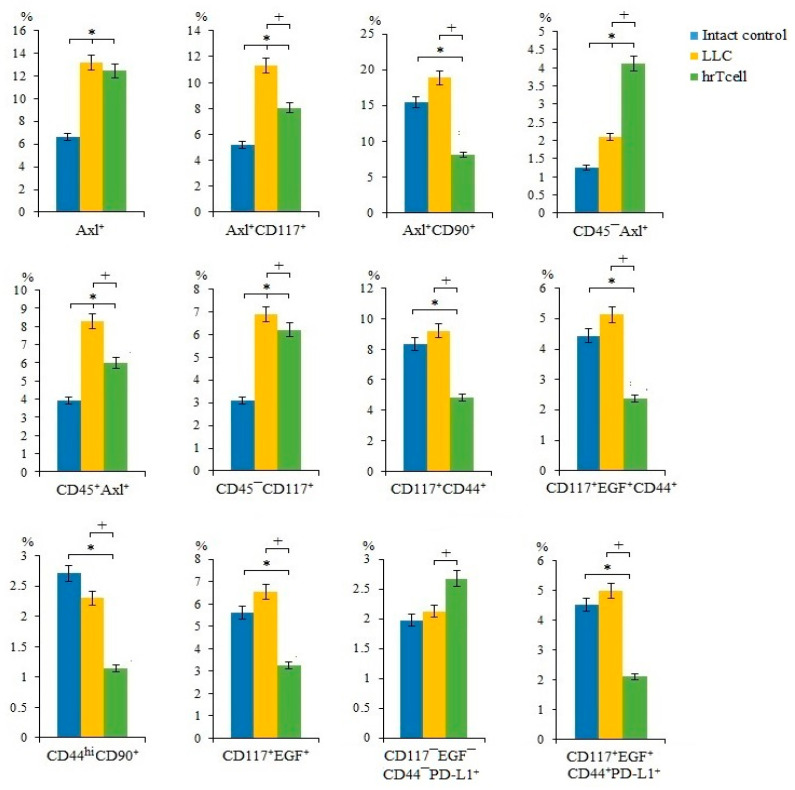
The effect of therapy with human reprogrammed CD8^+^ T-cells (hrT-cell) on the level of cancer cells and cancer stem cells (CSCs). The proportion (% of total mononuclear cells number) of Axl^+^**,** CD117^+^CD44^+^, CD117^+^EGF^+^CD44^+^, CD117^+^EGF^+^, CD117^-^EGF^-^CD44^-^PD-L1^+^, CD117^+^EGF^+^CD44^+^PD-L1^+^, Axl^+^CD117^+^, Axl^+^CD90^+^, CD45^+^Axl^+^, CD44^hi^CD90^+^, CD45^-^Axl^+^, and CD45^-^CD117^+^ CSCs in lungs of C57BL/6 mice with LLC on d17. Cells were analyzed by flow cytometry using antibodies for CD117, CD44, CD276, CD90, Axl, EGF, and PD-L1. * Differences are significant in comparison with the intact group (*p* < 0.05); +—for comparison with the mice with LLC (metastatic model). The results are presented from 3 independent series of experiments.

**Figure 4 ijms-23-15780-f004:**
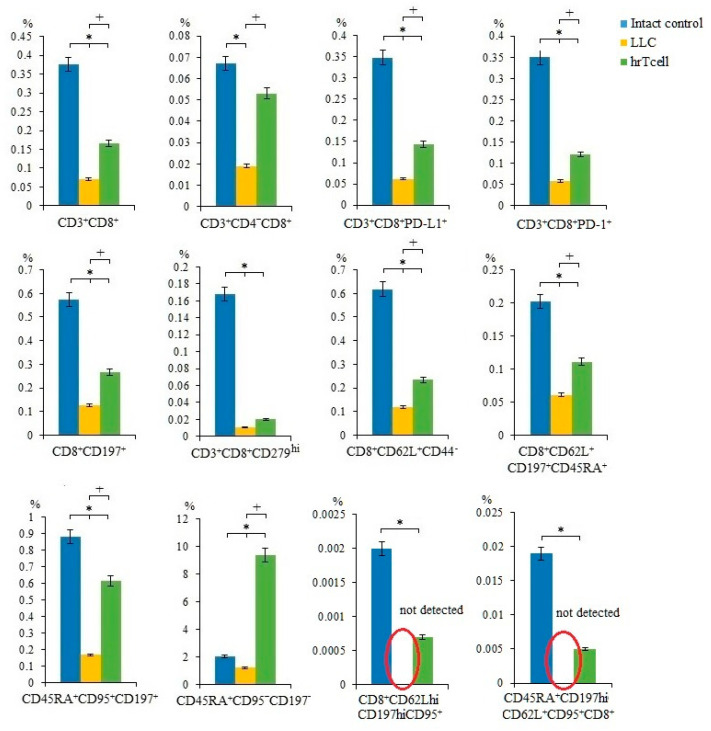
The effect of therapy with human reprogrammed CD8^+^ T-cells (hrT-cell) on level of T-cells in the blood. The proportion (% of total mononuclear cells number) of CD3^+^CD8^+^, CD3^+^CD4^-^CD8^+^, CD3^+^CD8^+^PD-L1^+^, CD3^+^CD8^+^PD-1^+^, CD8^+^CD197^+^, CD3^+^CD8^+^CD279^hi^, CD8^+^CD62L^+^CD44^-^, CD8^+^CD62L^+^CD197^+^CD45RA^+^, CD45RA^+^CD95^+^CD197^+^, CD45RA^+^CD95^+^CD197^-^, CD8^+^CD62L^hi^CD197^hi^CD95^+^, CD45RA^+^CD197^hi^CD62L^+^CD95^+^CD8^+^ T-cells in blood C57BL/6 mice with LLC on d17. Cells were analyzed by flow cytometry using antibodies for CD3, CD4, CD8, CD44, CD62L, CD45RA, CD95, CD197, PD-1, and PD-L1. * Differences are significant in comparison with the intact group (*p* < 0.05); +—for comparison with the mice with LLC (metastatic model) (*p* < 0.05). The results are presented from 3 independent series of experiments.

**Figure 5 ijms-23-15780-f005:**
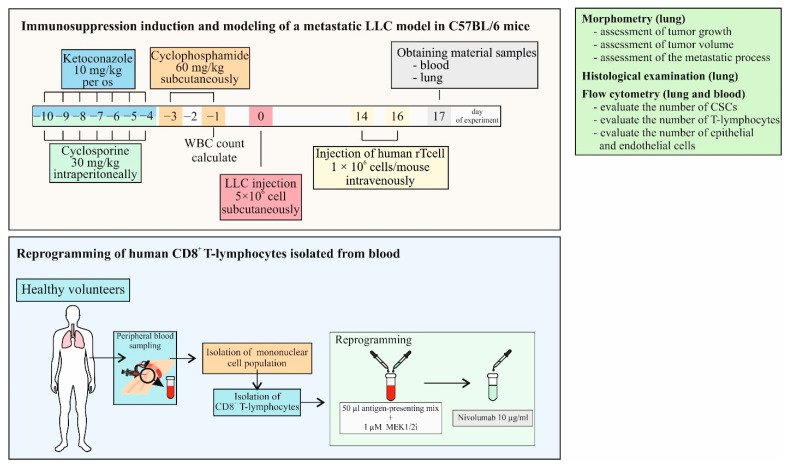
The experimental design of the study.

**Table 1 ijms-23-15780-t001:** Estimation of lung and tumor mass, tumor weight coefficient.

Group	Tumor Weight, mg	Tumor Weight Coefficient, mg/g
Intact	0	0
Mice with LLC	4.84 ± 0.57	175.62 ± 18.95
Mice with LLC + hrT-cells	5.09 ± 0.42	181.08 ± 13.54

**Table 2 ijms-23-15780-t002:** Assessment of tumor volume in animals.

Group	X, mm	Y, mm	Z, mm	V, mm^3^
Mice with LLC	27.33 ± 1.44	17.33 ± 1.33	12.78 ± 1.0 8	2899.60 ± 513.88
Mice with LLC + hrT-cells	27.44 ± 1.55	16.00 ± 0.67	14.33 ± 1.38	3055.4 ± 525.95

## Data Availability

Not applicable.

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
