# Peer review of "Cell Therapy with Human Reprogrammed CD8+ T-Cells Has Antimetastatic Effects on Lewis Lung Carcinoma in C57BL/6 Mice"

_ijms, 2022, doi:10.3390/ijms232415780_

Round 1

Reviewer 1 Report

Dear Authors,

The manuscript submitted is original and explains a novel method to improve T-cell therapy. My comments are :

1. Please explain the synonyms especially ones at line 38. 

2. in section 2.1.2, the tumor growth endpoint is shown as tumor volume and weight. Although it would be beneficial for readers to show baseline tumor burden before the intervention. 

3. The results associated with Fig 3 and 4 are not clearly explained in the main text. Please explain in the main text what are the cells expressing the mentioned markers. 

4. Cytotoxic activity of hrT-cells are not shown, which can be done by analyzing cytokine release either by ELISA or FACS. This is important to say that it is the cytotoxic effect of transferred hrT-cells.

5. The manuscript has many spelling and grammatical mistakes. Please correct those.

Author Response

We thank reviewer for their time and valuable comments. We now improved our manuscript. We have now revised the manuscript according to the suggestions. All changes have been included in the revised manuscript. Thank you for positive value of our paper.

Question 1

 Please explain the synonyms especially ones at line 38.

Answer

Thank you for this comment. We added transcription of abbreviations KRAS, EGFR and ALK in text.

Question 2

In section 2.1.2, the tumor growth endpoint is shown as tumor volume and weight. Although it would be beneficial for readers to show baseline tumor burden before the intervention.

Answer

Thank you for your comments. As a control of the baseline tumor burden, mice with LLC without treatment were used. Since inbred mice were used in the study, all animals were injected with the same number of LLC cells that animals with LLC without treatment can be as a control. Moreover, to calculate the indices, the ratio of tumor volume in control mice compared to treated mice at the same time was used [Hather G., Liu R., Bandi S., Mettetal J., Manfredi M., Shyu W.C., Donelan J., Chakravarty A. Growth rate analysis and efficient experimental design for tumor xenograft studies. Cancer Inform. 2014 Dec 9;13(Suppl 4):65-72. doi: 10.4137/CIN.S13974].

 Question 3

The results associated with Fig 3 and 4 are not clearly explained in the main text. Please explain in the main text what are the cells expressing the mentioned markers.

Answer

We added the information in our manuscript. Despite the fact that cancer stem cells (CSCs) from different tumors have been phenotypically and functionally characterized, there are still no specific markers that would clearly define CSC. CD44, CD87, CD117, CD276, Axl, EGF, and Ki67 are the most commonly used markers to identify CSCs in lung cancer [Maiuthed A., Chantarawong W., Chanvorachote, P. Lung Cancer Stem Cells and Cancer Stem Cell-targeting Natural Compounds. Anticancer Res. 2018, 38, 3797-3809. doi: 10.21873/anticanres]. Some selected markers were selected for this study. Currently, the identification of CSCs is challenging due to the lack of specific markers. Often, CD117, Axl, CD44, CD87, CD276, EGFR are used to identify CSCs.

Question 4

Cytotoxic activity of hrT-cells are not shown, which can be done by analyzing cytokine release either by ELISA or FACS. This is important to say that it is the cytotoxic effect of transferred hrT-cells.

Answer

We are grateful for your comment. Cytotoxicity of hrT-cell was evaluated in CSCs culture isolated from adhesive fraction of mononuclear cells from patient with SCLC. Cytotoxicity was assessed by analyzing the ratio of dead cells to the total number of CSCs. The cells were previously stained by Hoechst (to identify cell nuclei); 7 AAD (to identify dead cells). Cytation 5 Cell Imaging multi-mode reader (BioTek Instruments, Inc., Winooski, VT, USA) was used for analysis. Cytotoxicity data were added to supplement material (Figure S2).

 Question 5

The manuscript has many spelling and grammatical mistakes. Please correct those.

Answer

Thank you for the comment. We corrected spelling and grammatical mistakes.

Reviewer 2 Report

The manuscript by Skurikhin et al. describes  the effect of human reprogrammed CD8 positive T cells (hrT-cells) on Lewis Lung Carcinoma in C57BL/6 mice. The authors found that hrT- cells have anti- metastatic activity.

Figures 2-4: Histograms with the gating should be shown (examples).

Figure 5: Reprogramming….which MEK inhibitor was used?

Line 95: Fig.1 a,d should be replaced by b,e

How long was ketoconazole (line 309) and cyclosporine (line 312) administered? Should be mentioned in the text.

Author Response

We thank reviewer for their time and valuable comments. We now improved our manuscript. We have now revised the manuscript according to the suggestions. All changes have been included in the revised manuscript.

Question 1

Figures 2-4: Histograms with the gating should be shown (examples).

Answer

We now added histograms and dot-plots with the gating in supplement material (Figure S3 and Figure S4).

 Question 2

Figure 5: Reprogramming….which MEK inhibitor was used?

Answer

Thank you for the question. Monoclonal antibody nivolumab (Bristol-Myers Squibb Company, USA) and MEK 1/2 inhibitor (cat. PZ0162, Sigma-Aldrich, St. Louis, MO, USA) were used for reprogrammed. Information about the reagents that were used for reprogramming T-cells was added paragraph 4.11.

Question 3

Line 95: Fig.1 a,d should be replaced by b,e.

Answer

Thank you. We correct it.

Question 4

How long was ketoconazole (line 309) and cyclosporine (line 312) administered? Should be mentioned in the text.

Answer

Thank you for your question. Ketoconazole and cyclosporine were administrated during 7 days, from day 10 to day 4 prior of LLC injection. We added this information in paragraph 4.4.